# A Maturity Model Proposal for Industrial Maintenance and Its Application to the Railway Sector

Itxaro Errandonea [1,2,*] , Unai Alvarado [1,2], Sergio Beltrán [1,2,3] and Saioa Arrizabalaga [1,2,3]

1   CEIT-Basque Research and Technology Alliance (BRTA), Manuel Lardizabal 15,
    20018 Donostia/San Sebastián, Spain
2   School of Engineering, Universidad de Navarra, Tecnun, Manuel Lardizabal 13,
    20018 Donostia/San Sebastián, Spain
3   Institute of Data Science and Artificial Intelligence, DATAI, Universidad de Navarra, 31009 Pamplona, Spain
*   Correspondence: ierrandonea@ceit.es

**Abstract:** Maintenance is one of the major concerns of the industrial sector. Acquiring better levels of maintenance maturity is one of the objectives to be achieved. Therefore, prescriptive maintenance is one of the areas of recent research. Current works in literature are focused on specifics of maintenance strategies (from preventive to prescriptive), usually related to a fixed asset. No previous work has been identified regarding the methodology and guidelines to be followed to be able to evolve within the different strategies from a generic perspective. To address the lack of a methodology that shows a more evolutionary path between maintenance strategies, this paper presents Maintenance Maturity Model M3: a maturity model that identifies three areas of action, four levels of maturity, and the activities to be carried out in each of them to make progress in the maturity level of maintenance strategies. The implementation of prescriptive maintenance should be done in a gradual way, starting at the lowest levels. M3 approaches the problem from a broader perspective, analyzing the 18 different domains and the different levels of prior maturity to be considered for prescriptive maintenance. A study has also been carried out on the different maintenance actions and the applicability of the proposed M3 maturity model to the railway infrastructure maintenance is discussed. In addition, this paper also highlights future research lines and open issues.

**Keywords:** decision making; knowledge-based maintenance; knowledge support system; maturity model; prescriptive maintenance



## 1. Introduction

Maintenance costs are one of the main drivers of the operational expenses (OPEX) of companies, and are a very high dependent in the industrial sector [1]. Optimizing the execution of maintenance activities is key to reduce costs while maintaining the required quality standards. This is the reason why industries with higher maintenance costs (and those who are more mature in terms of technology deployment) are the ones that invest more in implementing more advanced and efficient maintenance strategies.

Gartner describes the path towards more advanced maintenance strategies, from preventive to prescriptive maintenance, reaching a higher level of maturity at each step [2,3]. As shown in Figure 1 starting with the basics of maintenance operations, tactics based on the knowledge and experience of maintenance managers are planned, enabling preventive maintenance. Eventually, with the help of different optimization algorithms, the logistics for better on-time use of resources can be improved. However, this strategy is not efficient as it leads to over-maintenance to guarantee service requirements.

The next step toward more efficient operations is the adoption of knowledge-intensive strategies such as condition-based maintenance (CBM). By incorporating status information of a given asset obtained from sensors, early detection of failures and diagnosis of different anomalies can be achieved. There are different approaches to be considered,

such as fault detection, the influence of disturbances, modelling errors, and the various uncertainties in today's systems [4–6]. However, there are still several gaps and challenges to overcome, starting with the incorporation of technologies such as IoT/IIoT (Internet of Things/Industrial Internet of Things) for asset monitoring. Other challenges must also be considered, for example, given the high volume of information to be managed in this type of system, the design of the data storage. It must also be considered that the data stored must be of high quality. Today, determining and enforcing the quality of the information is another challenge to be faced.

Once the actual condition of assets is addressed by means of CBM approaches, predictive maintenance adds an additional knowledge layer that is based on anticipated future conditions. To predict different failures affecting a component and the evolution of its condition over time, models that calculate its Remaining Useful Life (RUL), based on physical modelling, can be used. In lack of the necessary quantity or quality of data, data-driven models are not a viable solution. Moreover, they are models that can behave like a black box without providing deterministic information on their behavior. Physics-based models are better for this purpose; however, their high computational cost makes their applicability complex, depending on the use case. Functional alternatives would also be needed to overcome the lack of information, like hybrid models, which are also an alternative to the high computational costs of physical models.

Finally, optimal maintenance operations can be enabled by cognitive or prescriptive analytics, allowing us to fully automate processes based on predictive features, formalization of maintenance procedures and AI (Artificial Intelligence). The information on the procedures of the inspection of maintenance activities is necessary for the development of the decision-making model (decision models). Thanks to the advances made in terms of optimization, process definition and impact calculation, more reliable decision models can be obtained. However, there are still challenges to be addressed such as the definition of a detailed methodology that would benefit the development and deployment of prescriptive maintenance strategies.

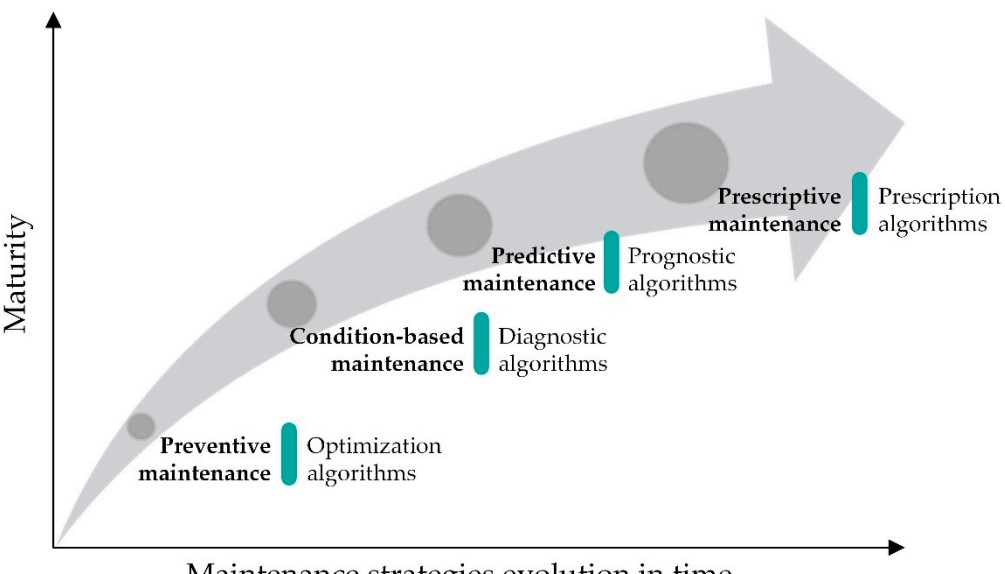

**Figure 1.** Knowledge-based maintenance evolution.

Hence, the implementation of prescriptive analytics is not immediate, and their deployment is usually done gradually, by first addressing CBM-based operations, and moving forward until prescriptive strategies are reached. It must also be taken into account that, if the asset of interest is not critical in production or does not affect safety, a prescriptive strategy may not be the most cost-efficient option [7].

To the authors' knowledge, no previous work has been found in the literature defining a roadmap for progressively improving maintenance strategies. It would be necessary to define the guidelines to be followed to acquire more advanced strategies, until reaching prescriptive maintenance. The approach presented in this paper is a maturity model: this defines the process to follow to reach the highest level of maturity within a discipline. The defined maturity model is named M3 (Maintenance Maturity Model) and has the purpose of defining a roadmap to navigate towards more advanced maintenance levels. To this end, this paper performs an analysis of the main domains identified as applicable within prescriptive maintenance strategies available in the literature, based on a defined methodology.

The rest of the article is organized as follows: Section 2 describes the research strategy and the methodology, including the selection of literature. In Section 3, the relevant domains within prescriptive maintenance are identified and maintenance maturity level analysis is carried out. Section 4 describes the maturity model proposed, followed by a discussion of a case in the railway domain in Section 5. Finally, conclusions are drawn in Section 6.

## 2. Material and Methods

Prescriptive maintenance is an emerging concept addressed by several research fields in recent years. There is limited work available in the scientific literature, as it is a relatively young term. Prescriptive maintenance is the last step in the adoption of an optimal maintenance strategy. Prescriptive maintenance consists of transforming preventive maintenance into a maintenance strategy based on a recommendation system for an optimized and knowledge-based activities plan [8]. This strategy begins with the collection of historical data as obtained in real time, to perform the diagnosis and prognosis of system failures. The objective is to optimize the scheduling of maintenance activities, with safety and cost-effectiveness.

To define the roadmap for adopting prescriptive maintenance strategies, a search strategy has been defined to identify the related work. A methodology followed for analyzing and extracting the necessary information from related work has also been defined and selected papers have been summarized.

### 2.1. Methodology Followed

The methodology used is composed of three phases (see Figure 2): the first was to select the relevant documents. The second was the application of a method to synthesize and extract the associated terms. Finally, the third phase was to perform an analysis between the results and the maintenance levels defined by Gartner.

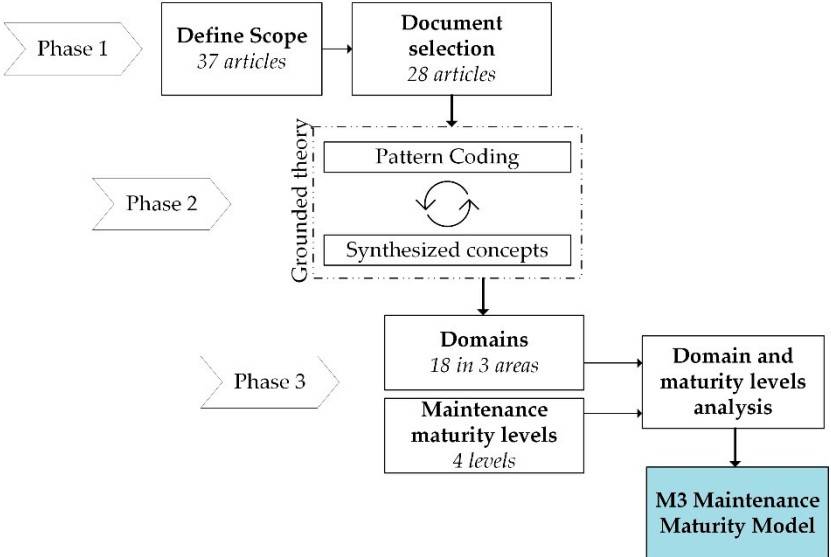

**Figure 2.** Methodology diagram.

Given the need to answer the questions raised, based on the selection criteria defined, 28 documents were obtained in the first phase. For the second phase, it was decided that methodologies based on the analysis of information found in documents were used. There are several defined methodologies that try to find patterns to synthesize information in bounded terms. They use iterative processes and identify common fundamentals [9,10].

The grounded theory methodology has been followed because it consists of two steps: the first one tries to detect common terms in different documents, while the second tries to group or synthesize terms into concepts or domains [11]. Several cases that have successfully used this methodology in different sectors can be found in the literature [12–14].

The maintenance point of view has been followed; therefore, all the analysis has been focused on the different areas and levels that define maintenance. Maintenance manages the information of what activity needs to be performed and when. In addition to this information, it should also manage what assets are to be maintained and why. The reason for the maintenance activity is normally based on the status of the asset. Taking this into account, three areas are defined for the domain analysis: assets, status and maintenance. The multiple analyses carried out during this work have been performed considering these three areas from different points of view, providing more information in each of them.

In the third phase, an analysis of the domains extracted from the second phase is carried out. On the one hand, an analysis has been performed in which the 18 domains are further divided into three areas: assets, status and maintenance. On the other hand, the four levels of maintenance defined by Gartner are reviewed to identify which levels correspond to each of the domains identified in phase 2. Finally, with the information extracted from the third phase, M3 is defined and presented: the Maintenance Maturity Model.

*2.2. Search Strategy*

To carry out the investigation of scientific works related to the concept of prescriptive maintenance, a search strategy has been defined. Certain questions were identified to focus the analysis on the works published so far by the scientific community. Next, literature searches were defined, determining search equations to be considered. Afterward, a series of acceptance criteria were defined to select the most relevant results.

The research process begun by identifying the specific questions to be answered. For this work, the following two research questions were raised:

- RQ1: What kind of methodologies or processes do maintainers follow for decision making in prescriptive maintenance?
- RQ2: Is there a framework that defines the steps to evolve and implement prescriptive maintenance strategies?

A related work analysis on the topic of prescriptive maintenance was carried out to select relevant documents that could contribute to the implementation of a maintenance decision making system. More precisely, the search was performed in Web of Science (WoS), one of the two major worldwide databases of bibliographic references and citations to periodic publications. Given that was not the intention to carry out an exhaustive review of the bibliography, only to search for related works, the terms used for this search were a combination of "prescriptive", "maintenance" and "decis*". Using the asterisk, several possible endings such as decision, decision-making, etc. were added. Given the need to find work related to all three terms at the same time, the logical and operator were used. Therefore, the search performed was "prescriptive AND maintenance AND decis*".

*2.3. Document Selection*

The search returned 37 results, but not all of them had the same relevance. The criteria for selecting a given document to be considered for further analysis were the following: (1) the document defines a prescriptive maintenance framework. (2) The document defines policies, actions and good practices for the implementation of prescriptive maintenance systems. Results that were not related to methods or solutions proposed for industry were

discarded. After applying the criteria defined above to the search results, 28 documents were selected and analyzed.

Table 1 contains the selected articles for further analysis. These include both articles published in Scientific Journals and articles presented at International Conferences. As it can be seen, the publications range from 1998 to 2020, and the frequency intensifies in current years. This gives an overview of the evolution of this area of study. Also presented is a summary of the results, where a broad range of use cases spanning different sectors are listed, from aerospace to manufacturing. The maintenance strategy tackled by each reference and its focus are also listed. Manufacturing is one of the most active industries in the research field of prescriptive maintenance. An analysis of the maturity level of the maintenance being studied was performed, and the process of implementation is described.

**Table 1.** List of the analyzed papers.

| N° | Ref. | Year | Title | Doc Type | Industry | Maintenance Strategy | Focus | Implementation Order |
|---|---|---|---|---|---|---|---|---|
| 1 | [15] | 2021 | Developing prescriptive maintenance strategies in the aviation industry based on a discrete-event simulation framework for post-prognostics decision making | Article | Aircraft | Prescriptive | Prescription | Yes |
| 2 | [16] | 2021 | Integrated Decision Support Systems (IDSS) for Dairy Farming: A Discussion on How to Improve Their Sustained Adoption | Article | Farming | Prescriptive | Decision support system | No |
| 3 | [17] | 2021 | Business analytics in Industry 4.0: A systematic review | Article | General | - | Maturity assessment | No |
| 4 | [18] | 2021 | A Two-Stage Data-Driven Spatiotemporal Analysis to Predict Failure Risk of Urban Sewer Systems Leveraging Machine Learning Algorithms | Article | Civil | Predictive | Data analysis + PHM | Yes |
| 5 | [19] | 2021 | Lifetime Benefit Analysis of Intelligent Maintenance: Simulation Modeling Approach and Industrial Case Study | Article | General | Predictive | Prescription | Yes |
| 6 | [20] | 2021 | A Machine Learning Approach to Enable Bulk Orders of Critical Spare-Parts in the Shipping Industry | Article | Navy | Prescriptive | Decision support system | Yes |
| 7 | [21] | 2021 | Concept and Economic Evaluation of Prescriptive Maintenance Strategies for an Automated Condition Monitoring System | Article | Aircraft | Prescriptive | Prognosis | Yes |
| 8 | [22] | 2020 | Predictive maintenance in the Industry 4.0: A systematic literature review | Article | Manufacturing | Predictive | Review | No |
| 9 | [23] | 2020 | Data-Driven Prescriptive Maintenance: Failure Prediction Using Ensemble Support Vector Classification for Optimal Process and Maintenance Scheduling | Article | Chemistry | Prescriptive | Prognosis | Yes |
| 10 | [24] | 2020 | Agile asset criticality assessment approach using decision-making grid | Article | Gas & Oil | Preventive | Asset criticality | Yes |
| 11 | [25] | 2020 | Developing flexible management strategies in infrastructure: The sequential expansion problem for infrastructure analysis (SEPIA) | Article | Construction | - | General | Yes |
| 12 | [26] | 2020 | Prescriptive Analytics for Swapping Aircraft Assignments at All Nippon Airways | Article | Aircraft | Prescriptive | Operational optimization | No |
| 13 | [27] | 2019 | A prognostic algorithm to prescribe improvement measures on throughput bottlenecks | Article | Manufacturing | Predictive | Prognosis + prescriptive | Yes |
| 14 | [28] | 2020 | Application of NARX neural network for predicting marine engine performance parameters | Article | Navy | Predictive | Prognosis | Yes |

**Table 1.** *Cont.*

| Nº | Ref. | Year | Title | Doc Type | Industry | Maintenance Strategy | Focus | Implementation Order |
|---|---|---|---|---|---|---|---|---|
| 15 | [3] | 2019 | PriMa: a prescriptive maintenance model for cyber-physical production systems | Article | Manufacturing | Prescriptive | Decision support system | Yes |
| 16 | [29] | 2019 | Prescriptive Maintenance of Railway Infrastructure: From Data Analytics to Decision Support | Conference | Railway | Prescriptive | Decision support system | Yes |
| 17 | [30] | 2018 | An asset-management oriented methodology for mine haul-fleet usage scheduling | Article | Mining | Prescriptive | Maintenance cost optimization | Yes |
| 18 | [31] | 2018 | Assessment of existing buildings performance using system dynamics technique | Article | Buildings | CBM | Diagnosis | No |
| 19 | [32] | 2018 | PriMa-X: A reference model for realizing prescriptive maintenance and assessing its maturity enhanced by machine learning | Conference | Manufacturing | Prescriptive | Maturity assessment | No |
| 20 | [33] | 2017 | Architecture for hybrid modelling and its application to diagnosis and prognosis with missing data | Article | Manufacturing | Prescriptive | Prognosis | Yes |
| 21 | [34] | 2016 | Big data analytics in logistics and supply chain management: Certain investigations for research and applications | Article | Manufacturing | - | Maturity framework | No |
| 22 | [35] | 2016 | Maintenance Analytics—The New Know in Maintenance | Conference | Railway | Prescriptive | Maintenance analytics concept | No |
| 23 | [36] | 2015 | Integrated approach to vessel energy efficiency | Article | Navy | - | Maintenance process flow | No |
| 24 | [37] | 2015 | Multiobjective and Multicriteria Problems and Decision Models | Book | General | Prescriptive | Decision model | No |
| 25 | [38] | 2013 | Predicting the behavior of solution alternatives within product improvement process | Conference | Manufacturing | Prescriptive | Decision process | No |
| 26 | [39] | 2002 | Effectiveness and cost benefits of computer-based decision aids for equipment maintenance | Article | Military | Prescriptive | Review | No |
| 27 | [40] | 1998 | An intelligent maintenance model (system): an application of the analytic hierarchy process and a fuzzy logic rule-based controller | Article | Manufacturing | Prescriptive | Decision model | - |
| 28 | [41] | 1998 | A decision-aid based on generalized multi-attribute utility for nuclear power plants maintenance | Conference | Nuclear plants | Prescriptive | Decision model | Yes |

The results obtained show that most of the works analyzed dealt with a specific problem, such as the diagnosis or prognosis of the health condition of the target asset. Those that address the problem in a more general way do not show guidelines for implementation; if they do, it is for a specific sector and a specific problem. Hence, it was necessary to carry out an identification of the main domains to be included in prescriptive maintenance strategies, and, furthermore, propose a guideline (maturity model) to navigate through the different steps towards the adoption of more advanced strategies enabling prescriptive capabilities.

## 3. Results of Domains and Maintenance Strategies Analysis

Through grounded theory methodology, the domains covered in the selected documents were extracted. The maintenance maturity levels defined by Gartner were also considered. In the following section, an analysis of this source material for the creation of

the Maintenance Maturity Model is carried out. To do this, the domains extracted were analyzed from two maintenance-related points of view: the three maintenance areas (asset, status and maintenance) and the four maturity levels of maintenance (preventive, CBM, predictive prescriptive).

### 3.1. Domains in Prescriptive Maintenance

Both RQ1 and RQ2 are oriented towards prescriptive maintenance. The search performed contained these two terms: "prescriptive" and "maintenance". Therefore, it can be stated that all the domains extracted from the selected documents can be found in the prescriptive maintenance strategy. The identified 18 domains have been classified into three defined areas: asset-oriented domains, asset status-oriented domains and maintenance-oriented domains. Figure 3 shows all the domains identified, and relevant references are also provided in each domain to those articles where more emphasis has been placed on the domain.

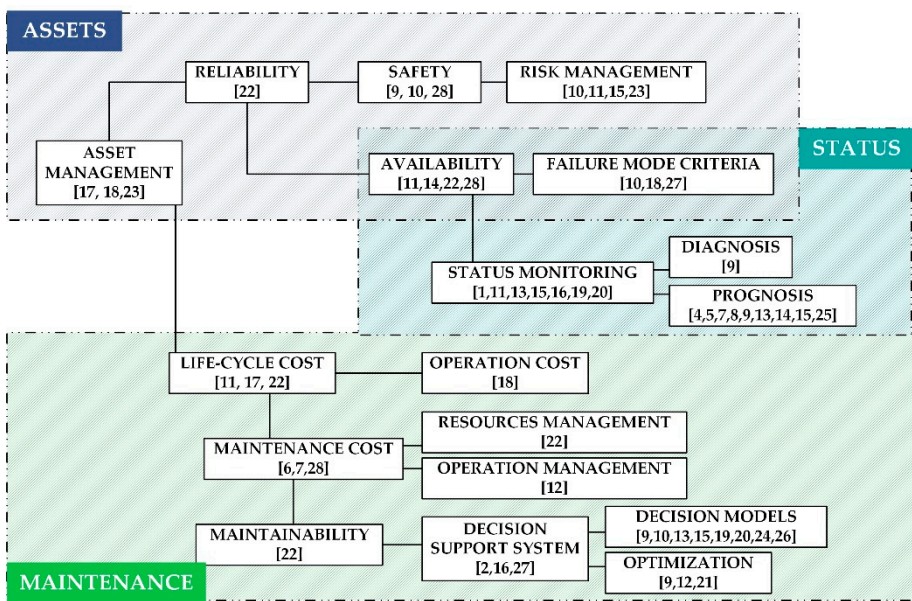

**Figure 3.** Prescriptive maintenance domains.

ASSET: The first area was considered to include all the domains that refer to the definition of the asset itself. This definition should include not only basic information on the manufacturing and installation of the asset, but also an analysis of its criticality. The risk management of the asset can then be evaluated by the level of safety required. Criticality can also be conditioned by the availability of the asset. The implications of the asset being out of service should be evaluated. These two points of view emerge in the reliability domain. To do this, it is necessary to consider the implications of an asset failing and/or becoming inoperative, defining failure modes. All the information analyzed for asset management was used as a basis for the analysis and management of the next two areas.

STATUS: This second area was considered for the domains describing the management of the health condition of assets. To guarantee the required level of availability of a given asset, its failure modes and the associated criteria to evaluate its current condition (e.g., by means of KPIs—Key Performance Indicators) must be defined. Once all failure modes and KPIs are identified, it is necessary to identify and define how the associated measurements to obtain such KPIs will be performed. In this way, discrete inspections of the asset were performed to evaluate its status.

Depending on the criticality of the asset, preventive activities may not be sufficient. Therefore, it is necessary to consider monitoring systems to evaluate performance in a more continuous way. There are also cases where monitoring systems need to be accompanied

by diagnostic systems. If the level of criticality is even higher, it would also enter the prognostic domain. In this way, we could anticipate a warning state in the asset.

MAINTENANCE: In this area there are domains linked to the maintenance and life cycle cost of the asset. Depending on the criticality of the asset, it can be necessary to adopt more advanced maintenance strategies and therefore address more complex domains. In all assets, regardless of the criticality, efficient management of the resources (personal, material, spatial and temporal) associated with operations and maintenance activities must be performed. Depending on the OPEX and criticality, domains such as decision support systems should be addressed together with decision and optimization models, thus studying the maintainability of the asset.

Apart from the three areas mentioned above, there was an additional domain identified in the analysis: maturity levels. Several research papers refer to maturity levels in maintenance. Using different terms, they analyze the evolution through different maintenance strategies. Several papers analyze in more detail case studies of the analytical maturity model levels proposed by Gartner [17]. However, there is no clear guidance on the assessment of the maturity level, not only for the adoption of prescriptive maintenance [3,32] but also for the rest of the strategies. It would be necessary to define the available technologies to assess the maturity level of the strategy that can be acquired [15].

The relationships and classifications described are those identified in the literature. It may be that there are more complex relationships between domains, however, the most evident are those reflected in the proposal made.

### 3.2. Maintenance Strategies

Figure 3 shows a possible definition of a process or methodology for prescriptive maintenance conditioned by 18 domains of study. Given Gartner's description of the evolution of strategies, it is to be expected that several of the identified domains belong to previous strategies to the prescriptive one.

Before discussing the relationship between the domains and the maintenance maturity levels, the definition of the different maintenance strategies is provided:

- Preventive maintenance: such a strategy is acquired to mitigate the consequences of reactive maintenance. Also known as time-based maintenance, this approach helps to prevent asset failures and breakdowns [42]. This type of strategy is based on the maintainer's knowledge of the asset's behavior in the event of failures to which it is subjected. As a result, periodic preventive planning of maintenance activities is obtained considering the necessary resources, the availability of the service and the impact on operations [43].
- Condition-based maintenance: also known as diagnosis-based maintenance, this strategy is based on the detection of anomalies by applying different technologies for asset monitoring and failure diagnosis [2]. With the information obtained from the diagnosis, it is possible to plan activities based on the current state of degradation of the asset, reducing the costs owed to over-maintenance [44].
- Predictive maintenance: also known as prognostic-based maintenance, this strategy performs maintenance scheduling based on observed future trends in asset behavior [45]. It uses available monitoring information, physics-based models, data analysis and uncertainty analysis by applying different techniques to anticipate the degradation of assets. Not only the level of degradation of the defect is taken into account but also the level of deterioration due to its operation conditions [19].
- Prescriptive maintenance: also known as knowledge-based or cognitive maintenance, this strategy is based on optimizing maintenance based on condition predictions by an AI (Artificial Intelligence) engine. The impact that this planning has on service availability, costs and safety is analyzed by decision models to optimize it [29].

When focusing on the maintenance of an asset, an issue needs to be raised before starting the definition of the strategy. It relates to the asset itself. The type of asset, and above

all, the impact the failure of the asset has, needs to be taken into account, as prescriptive maintenance strategies are more related to critical assets.

In summary, the deployment of preventive maintenance requires the definition of failure modes (FM), inspection reports, a well-defined and time-based plan of activities and a list of resources necessary to carry them out. For condition-based maintenance, the difference from the previous strategy is the addition of asset monitoring and fault diagnosis. In this way, maintenance activities are carried out based on the actual condition of assets. Predictive maintenance is based on obtaining a prognosis of the asset's condition. This way, maintenance activities can be scheduled accordingly, with the support of operational information. Finally, it would only remain to optimize the maintenance procedure and add decision models based on the information available to automate the execution of maintenance. Once the risk analysis, service unavailability and life cycle cost have been formalized, the automation process is more reliable.

It can be concluded that prescriptive maintenance is a difficult strategy to adopt in a single step. The following section describes a new solution, based on the evolutionary path that Gartner proposes for maintenance, to approach this strategy in an additive (constructive) way, focusing on each of the areas to be evolved (asset, asset status and maintenance) and providing guidance for their evolution as part of the so-called Maintenance Maturity Model (M3) proposal.

## 4. M3: Maintenance Maturity Model

A maturity model is a set of characteristics, attributes or indicators that represent progression in each discipline. In this case, a maturity model for maintenance is presented. This type of model allows for establishing a series of guidelines and references for the integration of processes or practices in industries. A maturity model facilitates the determination of the current level of maturity of a given paradigm and provides guidelines on how to improve the current situation towards more advanced scenarios [46].

There are three categories of maturity models: progression models, capability models and hybrid models [46]. Progression models are those that are specifically focused on defining attributes that assess maturity. Capability models define the capabilities required to reach the next maturity level. They measure the capabilities at each step of the progression. On the other hand, hybrid models are a mix of the two maturity models described above.

Maintenance Maturity Model (M3) is a hybrid maturity model (see Figure 4). This type of maturity model reflects the capabilities acquired at the different transitions between levels, and at the same time defines the attributes needed to follow the progression. This type of model is a roadmap for acquiring maturity. M3, contrary to the literature proposals, does not focus on any concrete maintenance strategy and provides guidelines or implementation order to make a maturity assessment of the current maintenance. It also provides the necessary guidelines to evolve to more advanced strategies.

Figure 4 shows that M3 is divided into four levels: preventive maintenance, condition-based maintenance, predictive maintenance and, finally, prescriptive maintenance. The characteristics or capabilities to be acquired at each of the levels are also divided into 3 areas: assets, status and maintenance. M3 is designed to acquire smarter maintenance strategies starting from the level on the left, in the right direction. At each level, guidelines described from top to bottom should be followed highlighting the new capabilities required.

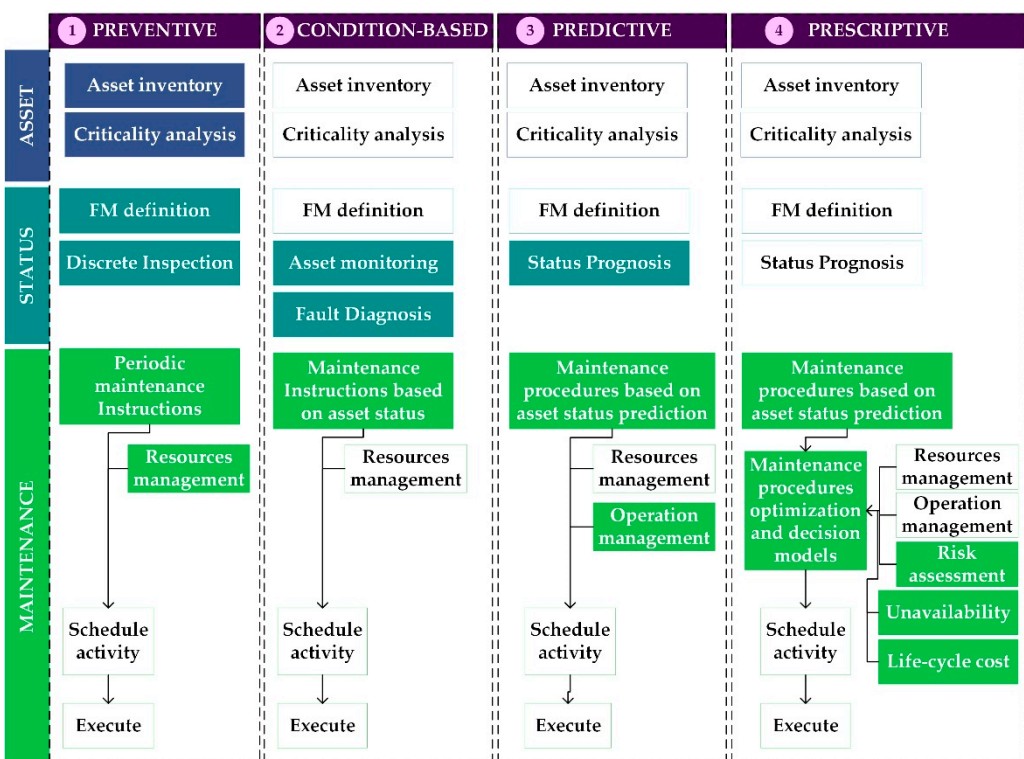

**Figure 4.** M3: Maintenance Maturity Model.

## 4.1. Assets

The initial focus is on the management of the asset itself. Basic information such as manufacturer, date of manufacture, date of installation, date of installation, its position or place in the infrastructure and the assigned identifier would be the first things to inventory and manage digitally. The digitalization of all the information related to the asset's life cycle is paramount. From the moment it is installed, until its replacement or elimination, with complete traceability of the maintenance that has been performed on it and all the information obtained, there is a reference point: the asset.

As previously mentioned, a criticality analysis should be performed. In this way, it is possible to define to what level of maturity it is cost-effective to move forward. In this analysis, the implications of the asset with respect to safety, availability and reliability must be studied. For that, all the risks must be analyzed and in this way, the criticality of the asset must be evaluated. In safety-critical environments, the requirements to be met are defined by different norms and standards. In each of the sectors, each focus of interest has a standard that describes how to fulfill acceptable safety levels.

## 4.2. Status

In the status area, the first activity to be performed is the definition of the failure modes of the inventoried asset. The failure modes of an asset have an impact on its safety and reliability. They need to be rigorously defined. This implies carrying out a study of the key performance indicators (KPIs) required for the assessment of the different failure modes.

When selecting the most critical failure modes and KPIs, methods such as FMECA (Failure Modes, Effects and Criticality Analysis) are often used [47–49]. This is a systematic analysis of potential failure modes of critical assets. It usually includes the identification of failure modes, their possible causes and the risk analysis associated with their effects and consequences. It also analyses the possible corrective and preventive actions to be taken that have an impact on the failure mode.

This analysis focuses on three parameters: Failure Severity (S), Failure Occurrence (O) and Failure Detection (O) in a range from 0 to 10. FMECA provides a quantitative way to calculate the Risk Priority Number (RPN).

$$RPN = S * O * D \tag{1}$$

The analysis performed results in a proposal of an innovation to minimize the severity of the failure, reduce the occurrence of the failure or improve the detection of the failure. The next step is based on quantifying the cost (C) of this innovation. The result obtained is identified as RCPI (Risk Cost Priority Index).

$$RCPI = S * O * D * C \tag{2}$$

This type of analysis helps to benchmark and determine what improvements should be made in terms of reliability, availability, maintainability and safety. FMECA provides identification of vulnerabilities and potential failures by capturing accurate information and engineering knowledge.

In this same area, reference is also made to the different attributes for achieving a better assessment of these failure modes and the detection of defects. For the assessment of the condition of the asset, at the most basic level of maintenance (preventive), there are discrete inspections performed with a defined frequency. If the level of criticality of the asset is increased, more intelligent forms of inspection are used, such as continuous monitoring (e.g., enabled by IoT technologies). By means of different sensors, more continuous measurements can be obtained over time. Sometimes these measurements do not give direct results of the evaluation of the defined KPIs, therefore, indirect diagnostic algorithms must be developed. For the predictive and prescriptive maintenance levels, it would only remain to include asset health prognostics practices. This can be addressed using techniques based on AI (Artificial Intelligence) technologies. Through these attributes, improvements in the availability and profitability of the managed assets are achieved. With the support of data acquisition, analysis, physics-based modelling and uncertainty calculation, among other things, it is possible to assess the health condition of an asset.

### 4.3. Maintenance

In the maintenance area, a flow chart is shown (see Figure 4 to define the guidelines to be followed. At the preventive level, the flow starts with periodic maintenance instructions. At this level, maintenance service providers usually have a set of criteria and preventive instructions defined. By following the instructions and managing the necessary resources to carry them out, the activities necessary for their execution can be scheduled.

For the condition-based maintenance level, maintenance instructions are based on the current state of the asset, since trusted, current and diagnostic data are available. By adding basic resource management, the necessary maintenance can be planned and executed. For the predictive maintenance level, the associated activities and operations would be based on the maintenance procedures defined upon predictions made about the future condition of the asset. At this point, it would be necessary to consider the management of operations for action planning.

Finally, to achieve a prescriptive level of maintenance, it is necessary to consider several domains such as safety, life-cycle costs (LCC) and maintainability of the asset. For this, combined with maintenance procedures based on predictions of asset status, there are several domains to consider: resource management and operations management (mentioned in previous levels), risk management, service unavailability management and LCC. Once these domains are studied, an optimization of the defined maintenance procedures can be performed by means of decision models. In this way, activities can be planned and executed following a prescriptive maintenance strategy applying decision support systems.

This maturity model is designed to guide maintainers in adopting smarter maintenance strategies. Furthermore, the proposal has been made in the form of a maturity model allowing step-by-step progress. To improve the implementation process for acquiring prescriptive maintenance strategies, different use cases on the needs of railway infrastructure maintenance are analyzed in the following section.

## 5. Discussion: Railway Infrastructure Maintenance

At European level, maintenance costs for rail infrastructure and vehicles are esti-mated at more than EUR 20 billion per year. The distribution of maintenance costs varies between countries and organizations, but according to [50], on the infrastructure side, track costs represent between 40 and 70% (where rail defects account for 30%) of the above values respectively. The above figures are similar for all infrastructure managers (IMs) and train operating companies, totaling more than 330 in the EU and 1500 organizations worldwide, and accounting for more than EUR 8 billion of recurrent costs in the EU.

RAMS (Reliability, Availability, Maintainability, Safety) analysis is a well-known tool for creating safe and robust systems. Its practices and methods are described in the standard [51]. Although traditionally used with safety-related electronic systems, the use of RAMS has not been very popular when supporting the management of mechanical systems, such as railway infrastructure until recently [52].

The SUSTRAIL project [53] has conducted an FMECA analysis resulting in a ranking of the different identified failure modes suffered by railway infrastructure [54]. This analysis is an example of how to identify critical assets and their fault modes and KPIs. This is a first step in M3 before going forward with the definition of the maintenance strategy. Amongst it, it is identified that one potential failure mode for the track is poor track geometry. This can be caused by deterioration of components and general geometry degradation due to traffic. The analysis defines the potential effects of this failure mode to be premature component failure, poor performance, high maintenance and derailment.

The IMs involved in the project decided to adopt several innovations to deal with defective track geometry. Specifically, these are the IMs Network Rails (UK) and ADIF (Spain). To minimize the severity of the failure, they proposed to improve the quality and maintenance of the installation and fix it before it reached unacceptable levels. To reduce the occurrence of such a failure mode, they suggested specific geo-grids and under-sleeper pads. Finally, to improve the detection of faulty track geometry, they proposed monitoring the geometry at the appropriate frequency by optimized maintenance scheduling using new methods for degradation prediction.

IMs are guided by common standards such as EN 13848. This standard defines the acceptance criteria for track geometry. It contains several parts defining design and maintenance requirements. Part five of this standard gives a detailed description of what the acceptance levels are for four different parameters for track geometry: defects on gauge, level, alignment and twist. For each of them, the standard defines certain limits that determine alertness, the need for intervention or the need for immediate action.

However, this standard does not determine how often you should inspect and evaluate the parameters to be considered. Nowadays, the railway sector makes use of very traditional preventive maintenance strategies, which are mainly based on periodic inspections and on the experience/knowledge of the maintenance staff [55]. Hence, IMs mainly have a preventive maintenance maturity level for the track geometry maintenance: discrete inspection activities are carried out at a frequency dictated by preventive procedures. These inspections can be either visual or carried out by an instrumented inspection vehicle. Maintenance activities, such as tamping, are also governed by preventive procedures. This takes into account the management of the necessary resources, e.g., tamping machine, driver and operators.

There are also some other examples showing a higher maturity level for the same asset and failure mode (track geometry). As determined by the M3 maturity model, the next condition-based maturity model implies more frequent track monitoring. There are several

examples dealing with both track monitoring and fault diagnosis in the literature. The proposal presented in [56] presents a solution based on machine learning classifier models for the diagnosis of irregularities such as track geometry. On the other hand, ref. [57] presents a review of the characteristics of IMUs for their concrete use in the monitoring of track geometry. Methods can also be found that deal with different problems with the data obtained, such as the need to align large amounts of track geometry data despite measurement errors and/or missing data [58].

With these approaches, new options for track geometry monitoring would be available, and it should be noted that CBM (together with proper, digitalized historical data collection and the use of approaches such as those mentioned above to address data quality issues) is an important step before progressing towards the maturity model. However, the objective is to determine what the appropriate frequency would be. To get closer to that goal, as proposed in M3 and the innovations proposed by SUSTRAIL IMs, new methods for degradation prediction (status prognosis) should be used. In the literature, one can find proposals such as those launched by [59] for predicting track geometry that have been tested on United States railway networks. There are also approaches such as those shown in [60] that test an artificial neural network model for track geometry prediction on Swedish railway networks. Among the parameters studied by the model, the standard deviation of longitudinal level is the one that determines the tamping planning, one of the most applied maintenance activities [61,62]. Hence, the maintenance procedure (tamping) is based on the prediction of the asset status (standard deviation of longitudinal level).

Considering the guidelines set by M3, the next level of maturity to acquire would be prescriptive maintenance. For this, it would be necessary to comply with the previous points, to obtain prediction of the state of the asset and thus be able to optimize maintenance procedures based on variables such as safety or cost. For example, ref. [63] shows a proposal to determine appropriate track geometry limits to minimize annual maintenance costs. There are also works such as [64], which presents a new approach for optimizing maintenance intervals, an innovation objective set by SUSTRAIL. This proposal is based on optimization based on costs, costs due to service unavailability and safety costs, using the value of avoided mortality to weigh the importance of safety. It would be interesting to be able to apply solutions such as the one proposed in [65] to support IMs in making maintenance decisions in terms of reliability, safety, availability and cost.

The track geometry analysis in this section is only one of the failure modes identified by participants in SUSTRAIL project. Other potential failure modes of concern to the IMs are rail rolling contact fatigue (RCF), erosion and slip-on embankments, rock-falls, switch rail wear, rail joint failure and missing or worn rail pads. The proposed M3 maturity model can systematically be applied to the different potential assets and failure modes and is identified as future work.

Many of the solutions presented in the literature focus on the results of the developed system. They generally provide theoretical information on the solution, and define in a generalist way its implementation in practical cases or experimental pilots. One of the future works detected is the need to make visible the challenges that occur when applying a solution in production. In most cases, the proposed solution must be readjusted to real-life needs, so that the performance and functionality of the proposed solution meet the objective set.

Another future work that has been detected is the need to use tools to prevail in the integration and interoperability between the different designed solutions. In this way, the interaction between the different modules implemented can be simplified.

## 6. Conclusions

The presented maturity model shows the potential of allowing a gradual adoption of more advanced maintenance strategies, starting from preventive strategies and ending with prescriptive strategies. The main conclusions of this work are the following:

1. The 18 domains extracted using grounded theory methodologies are present in each of the maintenance areas (asset, status and maintenance) and in the four levels of maintenance maturity analyzed (preventive, condition-based, predictive and prescriptive).
2. The criticality of the asset and the failure modes need to be analyzed first. The more mature strategies (predictive and prescriptive) are mainly suitable for critical assets.
3. Another key point to consider is to address the problem holistically. Once the maturity level is identified and the levels are targeted, M3 provides the actions to take. Each of these should not be approached as a standalone problem. The overall objective is to maintain the possibility of integration and interoperability of the different solutions that are already implemented.
4. M3 is a maturity model which addresses all maintenance strategies, not just a specific one, as most of the proposals in the literature do. It also provides a tool for the assessment of the maturity level, rather than providing concrete solutions for a specific strategy. Finally, it provides a roadmap for implementation.
5. The discussion carried out using the example of railway infrastructure maintenance demonstrates the applicability of the model presented in this paper. The criticality of the asset selected (track geometry) has a direct impact on safety, and acquiring the prescriptive maintenance level is the final objective. The different initiatives to improve and acquire higher levels of maintenance maturity have also been analyzed, although these techniques are not yet extensively used by IMs.
6. Considering that the model presents methods used in several sectors, M3 presents the possibility of being applied to another sector where critical assets are involved.

**Author Contributions:** Conceptualization, I.E., S.B. and S.A.; Formal analysis, I.E., U.A. and S.A.; Investigation, I.E., U.A. and S.A.; Methodology, I.E., S.B. and S.A.; Resources, I.E., U.A. and S.A.; Supervision, S.A.; Writing—original draft, I.E. and S.A.; Writing—review & editing, I.E., U.A., S.B. and S.A. All authors have read and agreed to the published version of the manuscript.

**Funding:** This research received no external funding.

**Institutional Review Board Statement:** Not applicable.

**Informed Consent Statement:** Not applicable.

**Data Availability Statement:** Not applicable.

**Conflicts of Interest:** The authors declare no conflict of interest.

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
