# Peer review of "A Maturity Model Proposal for Industrial Maintenance and Its Application to the Railway Sector"

_applsci, doi:10.3390/app12168229_

Round 1

Reviewer 1 Report

Abstract: Needs to add the main contributions come with this work mapping to the implementation into the case study. Also the abstract needs more description about the importance of taking railway infrastructure maintenance.

Introduction  and related works: The author gave a good background about prescriptive maintenance and I think Table 1 and merge with Table 2 so the related works can describe clearly with their differences and Table 1 alone is meaningless. 

M3 Analysis: Sections 3 and 4 describe the same subject and I think this section may give more detailed analysis includinig some statistical analysis to enforce the main work conclusion and then apply this analysis into the case study.

Case study: I support to use some statistical anaylsis especially with this type of research direction so as such work can be applicable.

Conclusions: Good conclusions and just need to refer to future works with this reseach approach.

Reviewer 2 Report

Reviewer Comments

The followings should be carefully addressed in the revision to be published in Journal of Applied Sciences.

1-      The authors should be followed the instruction of the Journal Applied Sciences parts and sections in this manuscript.

2-      Complete mathematic calculation model with all nomenclature missing

3-      The abstract needs more quantitative results. The abstract section is an important and powerful representation of the research. It is better that the results should be presented with the support of specified data. Please provide your contribution and work novelty.

4-      The authors should indicate this technique to enhance system performance. Also, the author should add more references that discuss the effect of using this technique. It is recommended that the authors carry out wide analysis and comparison with the state-of-the-art studies.

5-      Most tables and figures are highly needed improve the quality of all tables and figures.

6-      Add references for all equations.

7-      I would also expect to validate with two more experimental works available in the literature.

8-      The literature review must be improved. Please highlight in the literature review the differences between previous papers and your paper. Please clearly indicate the knowledge gap and prove that it is a really not analyzed area of the field. Please indicate new approach / new methods in a comparison to the existing investigations (literature review should be extended).

9-      Description of Experimental Methods. More quantitative information about the grid selection (which method was used.

10-  You need to add error analysis of your results and add the error bars in your graphs to indicate your accuracy measurements.

11-  Improve work justification.

12-  More quantitative conclusions should be presented. Please prepare additional comparisons, some percentage differences. There is a lack of quantitative conclusions which should contain main findings from the paper and highlight the new and high novelty and contribution of your work to the field.

13-  Present the mathematical equation of the boundary conditions and initial condition.

14-  I would also suggest including in the conclusion section but also in several other places in the manuscript discussion and comparison with findings from other authors with similar published research work.

15-  The conclusion section on lacks in summative conclusions. The main results, novelty and academic contributions should be emphasized in this section. Moreover, are the results obtained in this paper really applicable in other similar researches?

16-  In the discussion development, it is very important to emphasize points of agreement or disagreement between results in this work and others cited in references part of manuscript.

17-  Authors should discuss limitations of the current study and possible improvements for future directions/research works.

18-  Finally, I recommend the author to read through the whole text and correct it to make it more reader-friendly.

Reviewer 3 Report

The authors address the problem of describing different types of maintenance strategies for industrial assets.

After an interesting search of the scientific literature on the subject, some characteristics of the different strategies in the literature are defined. The case study and conclusions obtained are very general and reflect considerations well known in the literature. More depth on the case study and thus on the conclusions would be necessary for this to be a scientific article and not merely a review of the literature on the topic addressed.

Reviewer 4 Report

The manuscript, in its present form, contains several weaknesses. Adequate revisions to the following points should be undertaken in order to justify recommendation for publication.

1. The advantages of the proposed method of this paper should be more highlighted.

2. Present a qualitative and quantitative comparative analysis of the proposed scheme with its conventional counterpart.

3. Authors should argue their choice of the performance evaluation indicators.

4. The Introduction could be updated with recent reviews dedicated to the references related to the topic addressed, particularly on fault detection approaches, influence of disturbances, modeling errors, various uncertainties in the real systems. A relevant recent review are: Data-driven control of hydraulic servo actuator based on adaptive dynamic programming, Discrete and Continuous Dynamical Systems - Series S;  Event-driven NN adaptive fixed-time control for nonlinear systems with guaranteed performance, Journal of the Franklin Institute; Asynchronous Fault Detection Observer for 2-D Markov Jump Systems, IEEE Transactions on Cybernetics; It is necessary to comment what would be changed in this case and make relation with the papers on this topic in Introduction section, and in that way, point out other contemporary approaches and possibilities. I believe this would further strengthen the introduction and lend support to the methodology applied in general.

5. The rationale on the choice of the particular set of parameters should be explained with more details. Have the authors experimented with other sets of values? What are the sensitivities of these parameters on the results?

Round 2

Reviewer 2 Report

The authors addressed the comments satisfactorily. So, the manuscript can be considered for possible publication.

Reviewer 4 Report

The revised version has addressed all my concerns, no further comments.